# Application of Cotton Swab–Ag Composite as Flexible Surface-Enhanced Raman Scattering Substrate for DMMP Detection

**DOI:** 10.3390/molecules28020520

**Published:** 2023-01-05

**Authors:** Wen-Chien Huang, Hong-Ru Chen

**Affiliations:** Department of Chemical and Materials Engineering, Chung Cheng Institute of Technology, National Defense University, Taoyuan 33551, Taiwan

**Keywords:** SERS cotton swab, silver nanoparticles, versatile SERS substrate, chemical warfare agent simulant (DMMP)

## Abstract

It is both important and required to quickly and accurately detect chemical warfare agents, such as the highly toxic nerve agent sarin. Surface-enhanced Raman scattering (SERS) has received considerable attention due to its rapid results, high sensitivity, non-destructive data acquisition, and unique spectroscopic fingerprint. In this work, we successfully prepared SERS cotton swabs (CSs) for the detection of the sarin simulant agent dimethyl methyl phosphonate (DMMP) by anchoring N1-(3-trimethoxysilylpropyl) diethylenetriamine (ATS)/silver nanoparticle (AgNP) nanocomposites on CSs using ATS as the stabilizer and coupling agent. Simultaneously, the binding mode and reaction mechanics between the AgNP, ATS, and CS were confirmed by XPS. The modified CSs exhibited good uniformity, stability, and adsorption capability for SERS measurements, enabling the adsorption and detection of DMMP residue from an irregular surface via a simple swabbing process, with a detection limit of 1 g/L. The relative standard deviations (RSDs) of RSD_710_ = 5.6% had high reproducibility. In this research, the fabrication method could easily be extended to other cellulose compounds, such as natural fibers and paper. Furthermore, the versatile SERS CSs can be used for the on-site detection of DMMP, particularly in civil and defense applications, to guarantee food security and the health of the population.

## 1. Introduction

Nerve agents (NAs) strongly bind to the essential human enzyme acetylcholinesterase (AChE) and are highly toxic chemical weapons. Organophosphate NAs are employed as chemical warfare agents, the most toxic substances, including sarin, tabun, cyclosarin, and soman [1]. Intoxication with an organophosphorus compound typically results in incontinence, salivation, muscle twitching, and ultimately death. If sarin is absorbed through the skin, eyes, or respiratory tract of a person, death may occur within one to ten minutes [2].

Several detection techniques for liquid NAs have been developed, such as high-performance liquid chromatography [3,4], liquid chromatography with mass spectrometry [5,6], Fourier transform/infrared spectrometry [7,8], and nuclear magnetic resonance spectrometry [9]. Although these detection methods produce fairly reliable results, there are limitations, such as inconveniences with regards to transportation in the field and the requirement for special personnel for operation. Considering the situational urgency that chemical agents may cause, these methods are, therefore, not suitable for in situ measurements. As a result, there is still a need for highly sensitive, quick, robust, and dependable technology that is readily operable by relatively untrained first responders and homeland security operatives in the field.

Surface-enhanced Raman scattering (SERS) spectroscopy is one of the most effective detection tools with vast applications in homeland security. It offers some obvious advantages, including a fast response, high sensitivity, and fingerprint characteristics, over other spectroscopic methods. SERS spectroscopy has been frequently used in the fields of the detection of explosives [10,11,12] and environmental pollutants [13,14]. However, because organophosphorus NAs have only a weak interaction with highly SERS-activated noble metal substrates, they are hardly adsorbed on the substrates, or have only a very short residence time on them [15]. It is a challenge to effectively catch such molecules that weakly interact with metal substrates for enhanced Raman-based detection.

In this study, dimethyl methyl phosphonate (DMMP) was chosen as the sarin simulation agent. Two steps were used to prepare the SERS cotton swabs (CSs). First, the preparation procedure of N1-(3-trimethoxysilylpropyl) diethylenetriamine (ATS)-incorporated silver nanoparticles (AgNPs) included the complexing adsorption of Ag^+^ onto the amino groups. After that, an in situ chemical reduction method was employed using sodium borohydride as a reducing agent. The three amine groups of ATS present in the ATS/AgNP nanocomposites not only interact with the AgNPs, but also prevent them from agglomerating. Second, the presence of ATS in the ATS/AgNP nanocomposites can further form covalent bonds on the CS surface via the reactions of hydrolysis and condensation. The SERS CSs were of high flexibility with strong adsorption, owing to the anchoring of the ATS/AgNP nanocomposites on cotton fibers using ATS as the coupling agent. The versatile SERS CS composites have the advantages of a low cost and a simple method for SERS detection by wiping and adsorption from the uneven surface of the sample. This versatile SERS CS potentially represents a valuable method for the on-site detection of the sarin simulation agent.

## 2. Results and Discussion

### 2.1. Characterization of ATS/AgNPs/CS

Figure 1 shows the UV–Vis spectrum of the as-synthesized amine-stabilized AgNPs. The absorbance maxima appeared at 409 nm, which was due to the typical plasmonic absorption of AgNPs [16]. The AgNPs were obtained as stable yellow colloids (Figure 1, inset). The results indicate that the silver ions in the ATS solution were reduced to AgNPs after treatment with the NaBH_4_ solution for 1 to 72 h, and the quantity of AgNPs decreased with the duration of the reaction. The reaction of ATS/AgNPs tended to be stable after 24 h, so a 24-hour sample was obtained for a follow-up experiment. ATS was used as a stabilizer of the colloidal AgNPs, which remained stable for a long time without a modification of optical properties or the appearance of precipitates.

Figure 2 shows an XRD pattern expressing the crystallinity and structural features of the prepared ATS and ATS/AgNP composite. A broad peak centered at a 2θ value of around 20–30° was assigned to the amorphous nature of ATS. No other diffraction peaks were detected. The diffraction pattern of the ATS/AgNP nanocomposite showed sharp characteristic peaks at 2θ = 38.2°, 44.2°, 64.4°, and 77.5°, corresponding to the (111), (200), (220), and (311) crystalline planes of the face-centered cubic (fcc) crystalline structure of metallic silver, respectively (JCPDS file No. 04-0783) [17]. In addition, the characteristic peak of ATS was retained in the ATS/AgNP nanocomposite, which appeared at about 22.5° [18]. Figure 3a presents a TEM image of an ATS/AgNP nanocomposite with a size of about 20 nm. The Figure 3b inset shows the (111) plane orientations present in a single particle. A lattice image shows an interplanar spacing of 0.24 nm, corresponding to the (111) lattice plane of cubic Ag (JCPDS Card No. 04-0783). The value agreed well with the d values for the (111) planes of the AgNPs [19]. These results are in good agreement with our XRD results. Thus, it is evident that the black-colored particles of the ATS/AgNP nanocomposite observed on TEM were AgNPs [20,21]. In addition, energy-dispersive X-ray analysis was utilized to determine the chemical composition of the ATS/AgNP nanocomposite. From the energy-dispersive spectra (EDS) of the composites (Figure 3c), it could be clearly seen that C (13.7%), N (11.9%), O (45.8%), Si (15.4%), and Ag (13.2%) were the main components. Figure 3d shows the SAED pattern of the ATS/AgNPs. The pattern was indexed using digital Micrograph software and was found to agree well with the planes of silver. Therefore, the UV–Vis, XRD, TEM, and EDS results all indicate that Ag^+^ was adsorbed by the amine functional groups on ATS and successfully reduced to AgNPs by simple in situ chemical reduction with NaBH_4_.

SEM images elaborated upon the features of the CS surface. The surface topography of the CS is shown in Figure 4a–d, and it was very clean before the bonding of the ATS/AgNP nanocomposite. When the ATS/AgNP nanocomposite was assembled onto the surface of the CS fibers via the reactions of hydrolysis and condensation, as shown in Figure 4e–g, the surface became roughened, but the fiber diameter remained unchanged. Figure 4h shows that the AgNPs were randomly deposited on the entire surface of the CS [22]. According to the qualitative EDS analysis, the presence of C (67.51%), N (5.17%), O (21.14%), Si (0.57%), and Ag (5.61%), the major components of ATS and AgNPs, confirmed the presence of ATS and AgNPs in the flexible ATS/AgNPs/CS (Figure 4j).

### 2.2. ATS/AgNPs/CS Conjugate Bonding Study

In order to demonstrate the chemical states of the atoms in ATS, ATS/AgNPs, and ATS/AgNPs/CS, X-ray photoelectron spectroscopy (XPS) was employed for investigation. Compared with the pristine ATS, the appearance of Ag 3d proved that Ag^+^ was successfully dispersed in the ATS/AgNP solution by an in situ reduction in AgNPs. The high-resolution Ag 3d spectrum of ATS/AgNPs, as shown in Figure 5a, presented peaks at binding energies of 368.5 eV and 364.5 eV, which corresponded with Ag 3d_5/2_ and Ag 3d_3/2_, respectively. The 6 eV splitting of the 3D doublet, owing to spin–orbit coupling, further substantiated the presence of Ag element in the ATS/AgNP nanocomposite. The high content of silver element observed from the XPS spectra was in good agreement with the EDS results and SEM images, indicating that a large quantity of AgNPs was dispersed in the ATS solution. According to curve-fitting, Ag 3d_5/2_ of ATS/AgNPs had two peaks with binding energies of 368.5 and 368.9 eV, which corresponded with Ag^0^ and Ag-N, respectively [23,24]. The high-resolution spectrum of Ag 3d of ATS/AgNPs/CS showed that the spectrum could be fitted to six peaks. Besides the four peaks attributed to Ag^0^ (blue curves) and Ag-N bonds (pink curves), there were two new peaks (green curves) at 369.9 and 375.9 eV. As positively charged silver ions on the surface of the nanoparticles were bound to the anionic oxygen of the hydroxy group of the CS, the Ag 3d_5/2_ peak at 369.9 eV and the Ag 3d_3/2_ peak at 375.9 eV were both attributed to the Ag←O bond. The Ag 3D level presented a clear shift to higher binding energies, attributed to electron-donor behavior through the Ag surface.

A detailed look at the high-resolution spectrum of N 1 s (Figure 5b) revealed two clear peaks at 399.1 and 400.9 eV, corresponding to C-NH_2_ and C-NH, respectively. The N 1s spectrum for ATS/AgNPs presented four obvious characteristic binding energy peaks located at 399.1 eV, 399.7 eV, 400.9 eV, and 407.2 eV. It was noticeable that the C-NH_2_ feature peak of ATS/AgNPs at 399.7 eV apparently shifted to a higher binding energy and decreased in intensity, corresponding to the N 1s spectrum of ATS. This result indicates that the interaction forces of the functional groups with Ag^+^ were of the order -NH_2_ > -NH-. In the high-resolution scan of N 1s after the Ag induction, a new peak appeared at 399.1 eV, attributable to a -HN→Ag species [25,26]. Meanwhile, the additional peak at 407.2 eV corresponded to nitrate ions present in the ATS/AgNP solution [27,28]. The N 1s level presented a clear shift to higher binding energies, indicating electron-donor behavior through nitrogen.

Moreover, ATS and ATS/AgNPs exhibited three O 1s peaks (Figure 5c) at 530.9 eV, 532.1 eV, and 532.9 eV, which corresponded with Si-O-C, Si-O-Si, and Si-OH, respectively. It was noticeable that the Si-O-C feature peak of ATS/AgNPs at 530.9 eV apparently decreased in intensity, corresponding to the N 1s spectrum of ATS. Meanwhile, the feature peak for ATS/AgNPs showed increased intensities of Si-OH and Si-O-Si, which were attributed to the reactions of hydrolysis and condensation of ATS present in the ATS/AgNP solution. The high-resolution spectrum of O 1s of ATS/AgNPs/CS could be fitted with five peaks. There were three new peaks at 530.0 (green curves), 530.9 (orange curves), and 534.0 eV (light purple curves), corresponding to Ag←O-C, Si-O-C, and C-OH, respectively. The Ag←O bond was derived from the combination of positively charged silver ions on the nanoparticle surface with the anionic oxygen of the hydroxyl group of CS. The C-OH was derived from the hydroxyl group of CS, while the Si-O-C bond was derived from the reaction following condensation of the Si atom in the ATS/AgNP solution with the anionic oxygen of the hydroxyl group of CS.

ATS and ATS/AgNPs exhibited two Si 2p peaks, respectively (Figure 5d), with ATS peaks at 101.8 eV and 102.5 eV, corresponding to O-Si-C and O-Si-O, respectively. The O-Si-C feature peak of ATS/AgNPs at 102.4 eV apparently decreased in intensity, corresponding to the Si 2p spectrum of ATS. Meanwhile, the feature peak for ATS/AgNPs showed an increased intensity of O-Si-O, which was attributed to the reactions of hydrolysis and condensation of ATS present in the ATS/AgNP solution. The high-resolution spectrum of Si 2p of ATS/AgNPs/CS could be fitted with three peaks. There was a new peak at 101.1 eV (green curves), corresponding to O-Si-C. The O atom was derived from the hydroxyl group of CS.

From the above analysis, it is clear that there were remarkable differences in the N 1s peaks between ATS and ATS/AgNPs. The ATS protective agent contained -NH_2_ and -NH functional groups, which served to interact with AgNPs through chemisorption or physisorption. The results may indicate that the ATS/AgNP nanocomplex was formed through a coordinated covalent bond, because the nitrogen has a greater tendency to donate its pair of electrons to a metal [29].

### 2.3. Sensitivity of ATS/AgNPs/CS

For the SERS sensitivity evaluation of ATS/AgNPs/CS, DMMP was diluted and dripped onto the versatile SERS CS at concentrations ranging from 1 g/L to 35 g/L (Figure 6a). There was a prominent feature peak in the SERS spectrum of DMMP at 710 cm^−1^; this feature peak was assigned to a combined vibrational mode, including the symmetrical stretching of the two single P-O bonds and the P-C bond [30,31,32]. It was observed that the intensity of the prominent peak of DMMP at 710 cm^−1^ declined steadily with a decrease in DMMP concentration. SERS spectra were collected from three different points of the versatile SERS CS. Figure 6b shows the linear calibration curve of different DMMP concentrations. The linear regression equation was y = 628.6 + 246.6x, and the correlation coefficient (*R*^2^) of 0.9947 was of an approximately linear relation. It was observed that a good linear relationship existed between the areas and concentrations of the Raman peaks. The limit of detection for DMMP using the versatile SERS CSs was 1 g/L. In addition, Table 1 compares the analytical performances of various SERS sensors for the detection limit of DMMP, indicating that our SERS sensors were comparable with other SERS sensors reported in the literature.

### 2.4. Reproducibility of ATS/AgNPs/CS

In order to verify the SERS signal reproducibility, we selected 10 random locations on the surface of a versatile SERS CS from which to obtain SERS spectra, as shown in Figure 7a. The results revealed that the RSD value calculated from the SERS signal areas at the prominent peak of 710 cm^−1^ from different locations was 5.6% (Figure 7b). From the observed results, it was concluded that a great SERS signal reproducibility of the versatile SERS CS was achieved, with a relatively low RSD value. The main reasons for the good reproducibility were the simplicity of the fabrication method and the innate homogeneity of the cotton fibers.

### 2.5. Stability of ATS/AgNPs/CS

Furthermore, the SERS stability of the versatile SERS CSs against DMMP was also investigated. The Raman spectra of the recently prepared and modified CSs and the modified CSs stored for three months were compared. Although the characteristic peak intensity decreased with time, a good enhancement effect was still observed (Figure 8).

## 3. Experimental

### 3.1. Materials and Apparatus

ATS, sodium borohydride (NaBH_4_), Silver nitrate (AgNO_3_) and DMMP were procured from Sigma-Aldrich, Missouri, USA. All solutions were prepared with ultrapure water with a resistivity of no less than 18.2 MΩ·cm^−1^. CSs were obtained from Yuh Shiuans, New Taipei City, Taiwan. UV–visible absorption spectra were measured using a CT-8600 UV–Vis spectrophotometer (ChromTech, Bad Camberg, Germany). The morphologic details and energy-dispersive X-rays of the samples were obtained using a scanning electron microscope (SEM, JSM-7600F, JEOL, Tokyo, Japan). The crystallinity and structural features of the composite were measured by an X-ray diffractometer (D2 PHASER, Bruker, Karlsruhe, Germany). The morphologies and lattices of the samples were measured using a Tecnai F30 high-resolution transmission electron microscope (HRTEM, FEI, Noord-Brabant, Netherlands). The composition of the composites was measured using an ESCALAB 250 X-ray photoelectron spectroscopy (XPS) spectrometer (VG Scientific, East Grinstead, UK). The SERS spectra of the samples were recorded and analyzed using a TRIAX 550 Raman spectrometer (Horiba, Kyoto, Japan). The excitation wavelength was 532 nm. All Raman spectra were acquired with an integration time of 8 s.

### 3.2. Synthesis of ATS-Stabilized AgNPs

An aqueous solution of ATS (100 µL, 3 M) was added to a AgNO_3_ (10 mL, 0.01 M) solution with stirring at room temperature for 24 h to allow sufficient interaction between Ag^+^ and the amino groups. Then, the mixture was added to a NaBH_4_ (30 mL, 0.02 M) solution with stirring at room temperature for 24 h. The reaction quickly turned from colorless to brown in a few minutes. After 6 h, the final color changed to yellow. Specifically, the preparation procedure of ATS-incorporated AgNPs included the complexing adsorption of Ag^+^ onto the amino groups, followed by a simple in situ chemical reduction method using NaBH_4_ as the reducing agent. ATS stabilized the AgNPs in a colloidal state owing to the complexing adsorption of Ag^+^ onto the amino groups.

### 3.3. Fabrication of SERS Cotton Swabs

The versatile SERS CSs were made as follows. The CSs were soaked in 200 µL of ATS/AgNPs solution for 0.5 h. Then, the resulting wet samples were parched in an oven at 70 °C for 4 h to activate the cotton fibers. The above step was repeated 3 times. The resulting product was designated ATS/AgNPs/CS. Finally, the versatile SERS CSs were saved in zipper storage bags and dried in a moisture-proof box for future use (Figure 9).

### 3.4. Detection of DMMP Using ATS/AgNPs/CS

For sensitivity testing, eight different concentrations of DMMP standard solution (1, 5, 10, 15, 20, 25, 30 and 35 g/L) were prepared. For the experiment, 200 µL of DMMP solution at various concentrations was dropped on the versatile SERS CS substrate, then dried in the air. After the evaporation of the solvent, the SERS spectrum was recorded at three stochastic points selected on each SERS CS.

For the reproducibility test, 200 µL of a 5 g/L DMMP solution was dropped on the modified CS substrate, then dried in the air. On the versatile SERS CS substrate, ten random locations were chosen and averaged for the recording of SERS spectra.

For the stability test, the modified CSs were saved in zipper storage bags and dried in a moisture-proof box. The storage time was three months. Then, 200 µL of 5 g/L DMMP solution was dropped on the versatile SERS CS substrate, and the Raman spectroscopy performance was compared between the CSs before and after storage.

## 4. Conclusions

In this study, we successfully prepared versatile SERS CSs for the detection of the sarin simulation agent DMMP by anchoring ATS/AgNP nanocomposites on CSs using ATS as the stabilizer and coupling agent. The versatile SERS CSs had an excellent water absorption capacity and flexibility. The versatile SERS CSs exhibited a good SERS performance with a detection limit of 1 g/L for DMMP. The relative standard deviations (RSDs) of RSD_710_ = 5.6% had a high reproducibility. Simultaneously, we demonstrated the binding modes and reaction mechanics between ATS, AgNPs, and CS in an XPS spectrum. In addition, the versatile SERS CS composites had the advantages of a low cost and an easy method of SERS detection by wiping and adsorption from the surface of the sample. Finally, this study provided a new method of application for defense and environmental monitoring that enables the on-site detection of the sarin simulation agent.

## Figures and Tables

**Figure 1 molecules-28-00520-f001:**
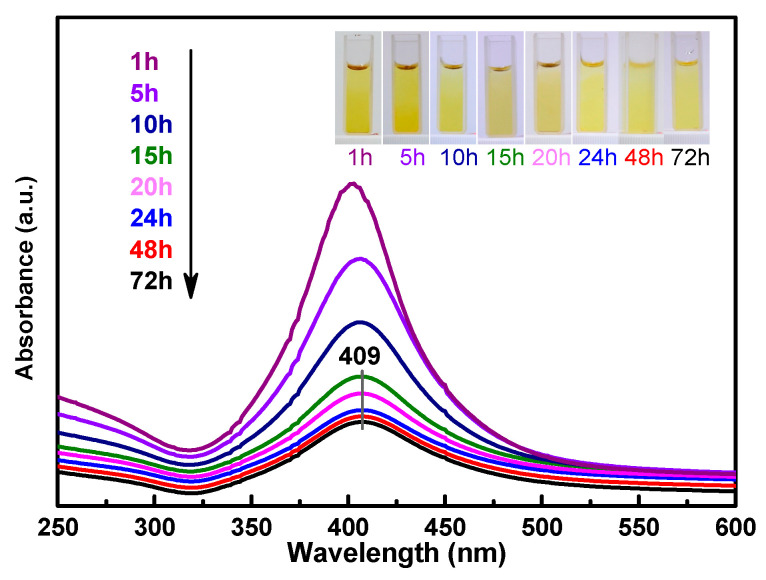
UV–Vis spectra indicating ATS/AgNPs synthesis recorded as a function of time (λ max = 409 nm). Inset images in UV–Vis spectra show colors ATS/AgNPs solutions.

**Figure 2 molecules-28-00520-f002:**
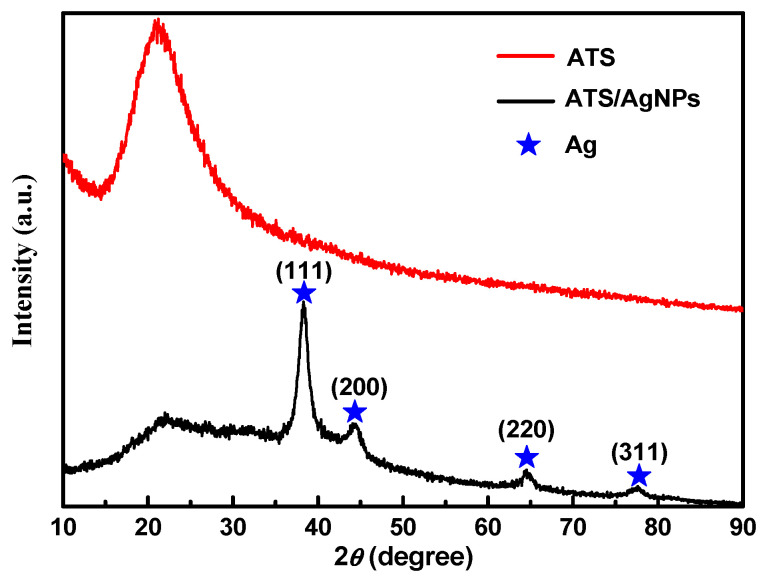
The XRD spectra of ATS and ATS/AgNPs.

**Figure 3 molecules-28-00520-f003:**
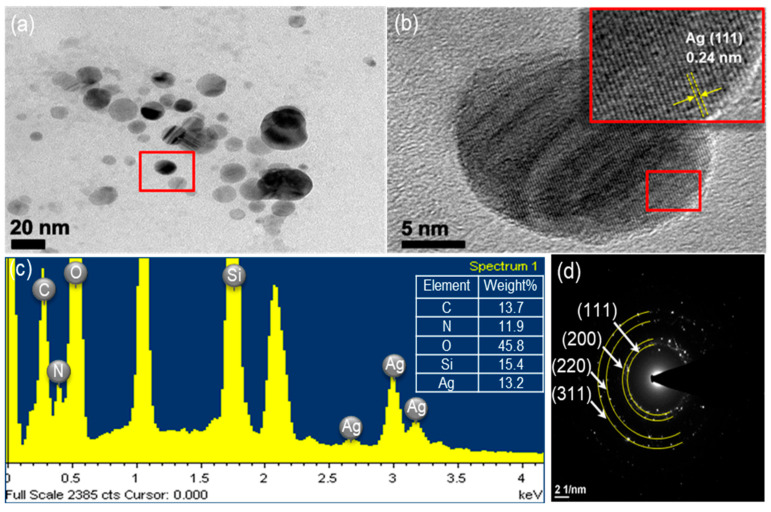
The ATS/AgNPs of (**a**) TEM images, (**b**) TEM images, (**c**) EDS spectrum and (**d**) SAED pattern.

**Figure 4 molecules-28-00520-f004:**
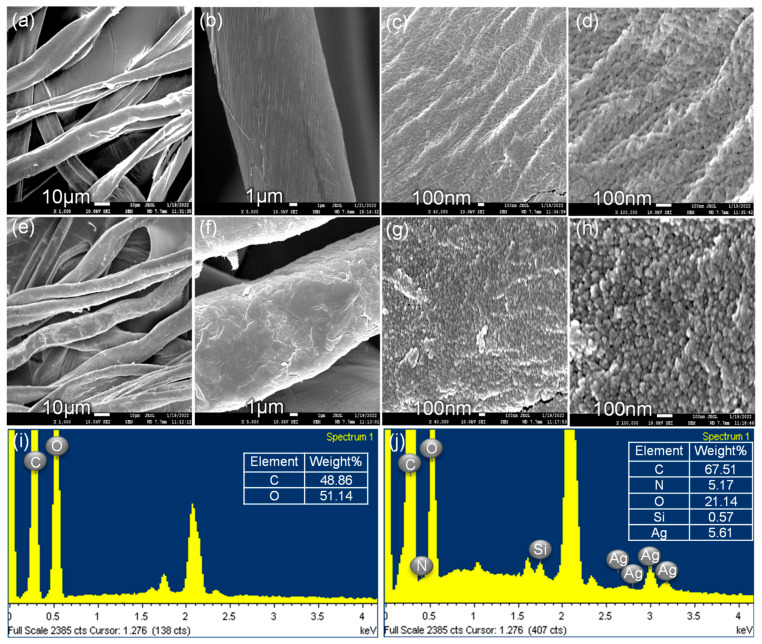
SEM images of the pristine CS (**a**–**d**) and the clusters of silver nanoparticles on CS surface (**e**–**h**). EDS spectrum of the pristine CS (**i**) and the clusters of silver nanoparticles on CS surface (**j**).

**Figure 5 molecules-28-00520-f005:**
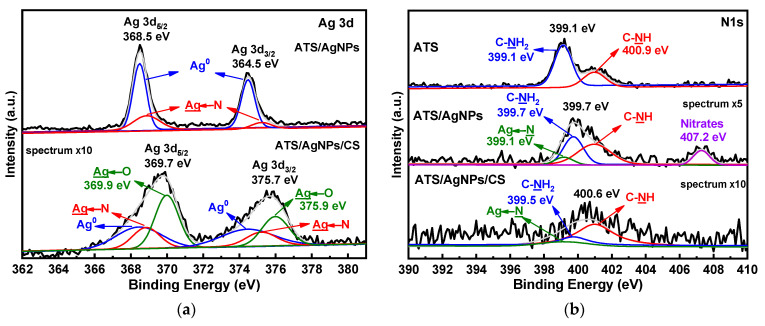
XPS spectra of ATS, ATS/AgNPs and ATS/AgNPs/CS in (**a**) Ag 3d, (**b**) N 1 s, (**c**) O 1 s, (**d**) Si 2p regions.

**Figure 6 molecules-28-00520-f006:**
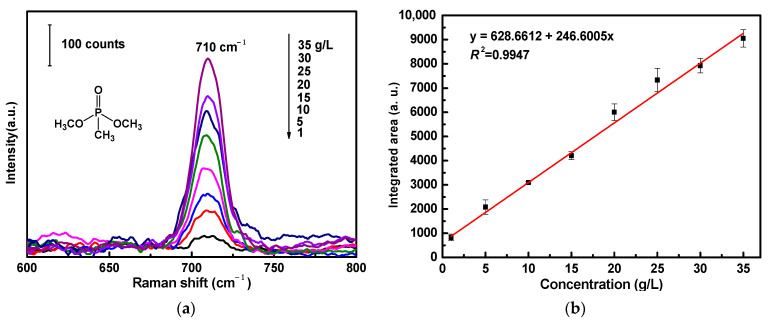
(**a**) SERS spectra of DMMP with varied concentrations (1 g/L to 35 g/L) acquired from ATS/AgNPs/CS. (**b**) The Raman intensity measured at 710 cm^−1^ was plotted as a function of DMMP concentrations.

**Figure 7 molecules-28-00520-f007:**
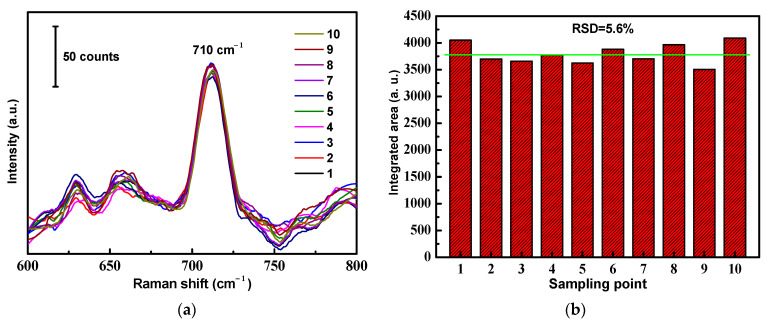
Reproducibility of the ATS/AgNPs/CS. (**a**) SERS spectra of DMMP (5 g/L) measured on ATS/AgNPs/CS from 10 random spots. (**b**) The band at 710 cm^−1^ was used for determination of the signal reproducibility of the substrate.

**Figure 8 molecules-28-00520-f008:**
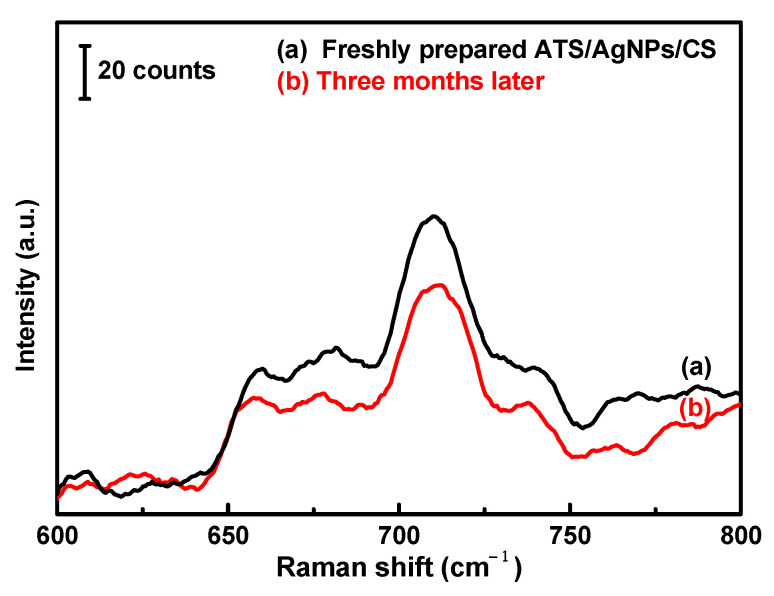
200 µL of 5 g/L DMMP solution was dropped on the versatile SERS cotton swab substrate. (a) Freshly prepared and (b) Three months later.

**Figure 9 molecules-28-00520-f009:**
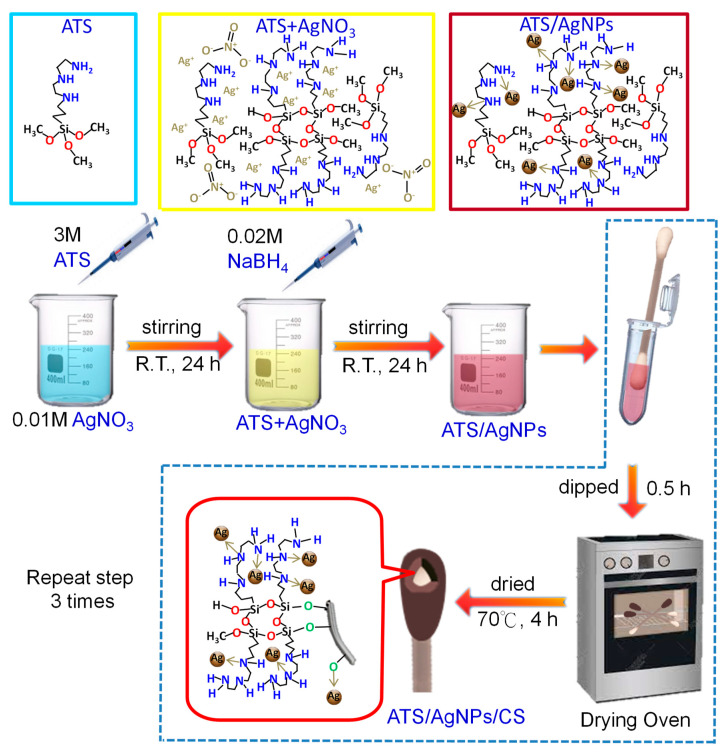
Schematic illustration of the fabrication process of the versatile SERS cotton swabs.

**Table 1 molecules-28-00520-t001:** The analytical performances of various SERS sensors for the detection limit of DMMP were compared.

Entry	SERS Substrates	Attribute	Sampling (Liquid)	Limit of Detection	Targeted Analytes	Ref.
1	AgO	Rigid	Drip Casting	1 ppm	DMMP CEES DEPA PMP	[33]
2	Au/Ag	Rigid	Drip Casting	~0.01 mg/m^3^	TPP DMMP Dicrotophos Malathion	[34]
3	PS/Au/ITO	Rigid	Drip Casting	0.1 mol/L	DMMP	[35]
4	Au@ZrO_2_	Rigid	Drip Casting	100 mg/kg	DMMP	[36]
5	ATS/AgNPs/CS	Flexible	Drip Casting Swabbing Adsorption	1 g/L	DMMP	This work

## Data Availability

All data related to this study are presented in this publication.

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
