# Peer review of "Application of Cotton Swab–Ag Composite as Flexible Surface-Enhanced Raman Scattering Substrate for DMMP Detection"

_molecules, 2023, doi:10.3390/molecules28020520_

Round 1

Reviewer 1 Report

In the submitted manuscript titled “Application of Cotton Swab-Ag Composite as Flexible Surface Enhanced Raman Scattering Substrate for DMMP Detection,” the authors present a development of a SERS-based sensor for dimethyl methyl phosphonate detection. After careful evaluation of the manuscript, I recommend the manuscript for publication after the following comments are addressed.

Please find my further comments below:

1)     In the introduction authors have mentioned that phosphorus NAs have only a weak interaction with highly SERS-activated noble metal substrates, they are hardly adsorbed on the substrates, or have only a concise residence time on them. How did the authors manage to adsorb it on the developed SERS substrate?

2)     During the preparation of the substrate, cotton swabs were heated at 70 degrees Celsius, what was the reason for applying heat? Was the heating done in a standard or inert atmosphere? If normal, what portion of Ag got converted to its oxide on heating in presence of atmospheric oxygen?

3)     Please provide EDX mapping of Ag-coated cotton swabs and calculate the percentage of Ag and O.

4)     Since the cotton swab by itself is very rough, I believe the arrangement of Ag nanoparticles will not be very uniform as a result, the SERS intensity for a particular concentration of DMMP should be different at random spots. However, from the spectra shown in figure 8a, the intensity of 10 different spots seems similar. Can authors explain this? At least 25 different spots should be included for analysis and a standard deviation must be calculated.

5)     Apart from sensitivity, selectivity is very important for a developed sensor. How do the authors suggest their SERS sensor will have selectivity towards DMMP in presence of other phosphate-containing molecules?

6)     A table showing the limit of detection achieved using other SERS-based sensors for DMMP can be included in the manuscript.

Author Response

Response to Reviewer 6 Comments

Point 1: In the introduction authors have mentioned that phosphorus NAs have only a weak interaction with highly SERS-activated noble metal substrates, they are hardly adsorbed on the substrates, or have only a concise residence time on them. How did the authors manage to adsorb it on the developed SERS substrate?

 Response 1: In this study, we choosed cotton swabs as SERS substrate. The cotton swabs were of distinguished flexibility and strong adsorption. The anchoring of ATS/AgNP nanocomposites on cotton fibers using ATS as the coupling agent. The SERS substrates after surface modification could selectively adsorb and enrich organophosphorus NAs on their surfaces. The interaction probability of phosphorus NAs and highly SERS-activated noble metal substrates  was increased. At the same time,  the residence time of phosphorus NAs  on the substrate was prolong.

Point 2: During the preparation of the substrate, cotton swabs were heated at 70 degrees Celsius, what was the reason for applying heat? Was the heating done in a standard or inert atmosphere? If normal, what portion of Ag got converted to its oxide on heating in presence of atmospheric oxygen?

Response 2:

(1)The resulting wet ATS/AgNPs/CS were parched in an oven at 70℃ for 4 h to activate the cotton fibers.

(2)The heating was done in a standard atmosphere.

(3)The three amine groups of ATS present in the ATS/AgNPs nanocomposites not only interact with the AgNPs, but also prevent them from agglomerating. Second, the presence of ATS in the ATS/AgNPs nanocomposites can further form covalent bonds on the cotton swab surface via the reactions of hydrolysis and condensation. Third, ATS as the stabilizer agent that prevent AgNPs got converted to oxide on heating in presence of atmospheric oxygen.

Point 3: Please provide EDX mapping of Ag-coated cotton swabs and calculate the percentage of Ag and O.

Response 3: According to the reviewer’s comment, we have added the graphs of EDS spectrum (Fig. 5 (i), (j)), and modified the statement on page 9, line 5. " According to the qualitative EDX analysis, the presence of C (67.51 %), N (5.17 %), O (21.14 %), Si (0.57 %), and Ag (5.61 %), the major components of ATS and AgNPs, confirmed the presence of ATS and AgNPs in the flexible ATS/AgNPs/CS (Figure 5 (j))."

Point 4: Since the cotton swab by itself is very rough, I believe the arrangement of Ag nanoparticles will not be very uniform as a result, the SERS intensity for a particular concentration of DMMP should be different at random spots. However, from the spectra shown in figure 8a, the intensity of 10 different spots seems similar. Can authors explain this? At least 25 different spots should be included for analysis and a standard deviation must be calculated.

Response 4:

(1)The surface topography of the cotton swab is shown in Figure 5(d). The surface of the cotton swab has holes before ATS/AgNPs nanocomposite bonding.

(2)When the ATS/AgNP nanocomposite was assembled onto the surface of the cotton swab fibers via the reactions of hydrolysis and condensation, as shown in Figure 5(h), the pores on the surface of the cotton swab are filled with uniform distribution of silver particles.

(3)Silver nanoparticles were uniform deposited  on the surface of the cotton swab.  These results were in good agreement with our Figure 8 results. The intensity of 10 different spots seems similar.

(4)The relative standard deviations (RSDs) of RSD710 = 5.6%,

Point 5: Apart from sensitivity, selectivity is very important for a developed sensor. How do the authors suggest their SERS sensor will have selectivity towards DMMP in presence of other phosphate- containing molecules?

Response 5: As the reviewer stated, apart from sensitivity, selectivity is very important for a developed sensor. In order to distinguish among the different phosphate molecules in the analyte, we will continue to use the ATS/AgNPs/CS to detect different types of phosphate molecules and build a database. The database will assist us in improving the accuracy of DMMP in the presence of other phosphate-containing molecules in the analyte.

Point 6: A table showing the limit of detection achieved using other SERS-based sensors for DMMP can be included in the manuscript.

Response 6:

(1)According to the reviewer’s comment, we have added the comparisons of the analytical performances of various SERS sensors for the detection limit of DMMP. (Table 1 )

(2)According to the reviewer’s comment, we have added the statement on page 12, line 4 from the bottom. "In addition, Table 1 compares the analytical performances of various SERS sensors for the detection limit of DMMP, indicating that our SERS sensors were comparable with other SERS sensors reported in literatures."

Reviewer 2 Report

Well done on a very good paper. I do not have any comments except the abstract can be further improved to emphasize the importance of the findings

Author Response

Response to Reviewer 2 Comments

Point 1: Well done on a very good paper. I do not have any comments except the abstract can be further improved to emphasize the importance of the findings.

Response 1: According to the reviewer’s comment, we have modified the statement on page 2, line 15. " In this research, the fabrication method could be easily extended to other cellulose compounds, such as natural fibers and paper. Furthermore, the versatile SERS CS can be used for on-site detection of DMMP, particularly in civil and defense applications, to guarantee food security and the health of the population."

Point 2: The manuscript should undergo extensive English revisions.

Response 2: According to the reviewer’s comment, we has been copyedited the manuscript by a native English-speaking professional copyeditor to a level. Certificate of English Editing is also included.

Certificate of English Editing (Please see the attachment.)

Round 2

Reviewer 1 Report

The authors have addressed all my comments and the manuscript can be considered for publication.